# Are We Getting Any Better? A Critical Analysis of Selected Healthy People 2020 Oral Health Indicators in 1999–2004 and 2013–2016, USA

**DOI:** 10.3390/ijerph19095250

**Published:** 2022-04-26

**Authors:** Carolina Marques Borges, Meghna Krishnamurthy

**Affiliations:** Department of Public Health, The College of New Jersey, Ewing, NJ 08618, USA

**Keywords:** oral health, Healthy People 2020, dental public health, oral epidemiology, disparities, social determinants

## Abstract

Oral health disparities are prevalent in the American population and are influenced by various social determinants. This study aimed to analyze oral health disparities in the US between 1999–2004 and 2013–2016 according to sociodemographic characteristics. This analytic cross-sectional study analyzed five oral health indicators from Healthy People 2020. A binomial test was used to compare proportions between baseline and follow-up. Only the indicator for non-treated cavities among children reached its goal. White children had the greatest decrease (−15.4%; *p* = 0.0428) in dental caries. Higher income determined better outcomes for adolescents (−27.54%; *p* = 0.00032 dental caries) and adults (−15.96%; *p* = 0.0143 tooth extractions). However, adults 35–44 years with the highest income had a significant increase (40.74%, *p* = 0.0258) in decayed teeth. This study provides evidence to suggest that some progress has been made towards reducing oral health disparities in the US, primarily among children. However, trends for certain indicators remain disparate between different racial/ethnic and income groups. Applications for the findings of this study should address the intersectional nature of social determinants of health and should center on improving the equity of services offered by public oral healthcare.

## 1. Introduction

Oral health inequities are unfair and unnecessary disparities in the oral health status of different population subgroups arising from structural issues in society and health systems. For example, according to the Centers for Disease Control and Prevention, the rate of untreated cavities in non-Hispanic Black or Mexican American adults is two to three times that of non-Hispanic White adults [1]. Educational attainment is also associated with disparate oral health status, as adults with less than a high school education are almost three times as likely to have untreated cavities as adults with some college education [1]. Socioeconomic differences are another determinant that translates into disparate health access and suboptimal health outcomes, as children aged 6 to 19 years from low-income households are 15% less likely to receive sealants and twice as likely to have untreated cavities compared to those from higher-income households [1]. The 2000 U.S. Surgeon General’s Report on Oral Health identified barriers to receipt of proper oral health care, including limited household income, transportation, flexibility in family schedule to make appointments, and lack of insurance [2]. The 2021 follow-up to the 2000 U.S. Surgeon General’s report, Oral Health in America, characterizes oral health as a global “tipping point” in that technological advancements and increasing consumer demands for aesthetic dental procedures have changed perceptions of dental care, exacerbating disparities for populations that continue to face barriers to essential oral healthcare [3]. These limitations to the adequate oral health care of Americans are concerning because oral health has implications beyond the maintenance of oral hygiene. It is a metric of overall health, without which everyday needs would be unfulfilled [4]. To address these basic health needs, it is essential for policies to recognize and reform the systems that allow marginalized populations to continue receiving suboptimal oral healthcare and education. Doing so first requires identifying the communities that are most impacted by oral health inequities, evaluating previous progress, and assessing their present health status so that future policies can effectively address their needs.

Healthy People 2020 is a federal initiative to improve the health of Americans, with particular attention paid to preventative health measures. Healthy People’s Oral Health topic area has 17 objectives addressing the prevention of craniofacial diseases and injuries as well as the improvement of preventative dental services and oral health education [4]. Healthy People’s demographic data on oral health topic areas have resulted in past successes, such as fluoridation of public water supplies for 7 out of 10 Americans [4]. The initiative also identifies objectives in need of improvement, including the development of community-based preventative health programs. Longitudinal data allow assessment of the progress made within each aspect of oral health, which can inform initiatives to bring Americans closer to achieving effective use of oral health care programs across communities. 

Healthy People 2020′s data also characterize the demographics of populations facing oral health disparities. The complete data reflect the progress of the United States population longitudinally with respect to social determinants including education level, total family income, race, sex, and comorbidity of oral health diseases with other diseases. Oral health trends based on social determinants are important in informing policies that can address disparities. The CDC broadly identifies overall trends in oral health as being the poorest among non-Hispanic Black, Hispanic, and Mexican Americans in the United States [1]. Yet, the longitudinal findings reported in *Oral Health in America* implicate more nuanced trends in oral health status of different populations, depending on a variety of factors including demographic qualities as well as the actual oral health metrics themselves. For instance, the disparity in dental caries prevalence between Mexican American and non-Hispanic White adolescents is reported to have increased in the past 20 years, while the gap between non-Hispanic Black and non-Hispanic White adolescents has decreased [3]. Among children aged 3–5 years, race and poverty concurrently influence dental caries experience [3,5]. Regarding other oral health metrics, such as edentulism in older adults, Black individuals experience nearly double the prevalence of edentulism relative to Mexican Americans and non-Hispanic White adults [3,6]. The findings can be further nuanced based on the breadth of the sampling frame. For instance, the prevalence of dental sealants among non-Hispanic Black and Mexican American adolescents is reported to have seen considerable progress broadly over the past two decades [3]. However, evaluations of the same indicators from 2011–2012 suggest that dental sealant prevalence was significantly lower among non-Hispanic Black adolescents compared to Hispanic, non-Hispanic White, and Asian adolescents [7]. The nuances of prior reporting on disparities in oral health trends therefore implicate certain important considerations for present studies, including the variety of demographics considered as well as the breadth and recency of the sampling frame. Having up-to-date data to understand trends is necessary to inform policy changes that can improve the oral health of specific vulnerable population subgroups. 

This study aimed to analyze oral health disparities in the United States, measured by selected oral health indicators from Healthy People 2020, between a baseline (1999–2004) and follow-up (2013–2016), according to sociodemographic characteristics.

## 2. Materials and Methods

This analytic cross-sectional study utilized secondary data from the open-access databases Healthy People 2020 and U.S. Census. Data were collected in January 2020 and assessed two periods: (1) baseline period from 1999 to 2004 and (2) follow-up period from 2013 to 2016. These time periods were selected based on completeness and availability of the most recent data. Demographic categories used included sex (male, female), race, and ethnicity (Hispanic or Latino, non-Hispanic or Latino, Mexican American, Black or African-American, White), income as a percentage of the federal poverty guideline (<100%, 100–199%, 200–399%, 400–499%, 500%+), country of birth (US, non-US), and insurance status (insured, private insurance, uninsured, public insurance).

Five out of 33 Healthy People 2020 oral health indicators were selected based on completeness of the available data. Each indicator that utilized a proportion estimate had a numerator and denominator component specific to the person, place, and time attributes of that indicator. Prevalence estimates, standard error values, and 95% confidence intervals for each sociodemographic variable were tabulated for each indicator. A complete description of each indicator used is provided in Table 1.

### 2.1. Descriptive Analysis 

The Healthy People 2020 datasets were used to obtain prevalence estimates as well as the standard error and 95% confidence interval for both time frames of each oral health indicator. We performed a descriptive analysis and determined percentage change between the two periods of observation. We calculated the relative change of the indicators in percentage terms by subtracting the estimate observed on period 2 by the estimate observed on period 1, then divided by the estimate observed on period 1. Final result was multiplied by 100 to be presented as a percentage. 

### 2.2. Binomial Proportion Test

The binomial test of proportions was used to assess whether there was a significant difference in independent sample proportions between the census groups from the baseline and follow-up periods. We assumed that each sample had a Bernoulli parameter distribution with the proportion of interest. Since it was a good estimator for the mean, we used an approximation by the normal distribution, as follows:

The null hypothesis (H_0_) assumed no difference between sample i (baseline) and sample j (follow-up) in regard to the proportion of children/adolescents with dental caries. The alternative hypothesis (H_1_) assumed that the proportion of children/adolescents with dental caries in sample i was different from (greater than or equal to) that of sample j. 

The same hypotheses were considered to test the equality of proportions between the adult group. A 5% significance level was used, allowing rejection of the null hypothesis for *p* < 0.05. GNU R version 3.6.1 was used to perform calculations.

## 3. Results

Of the five indicators analyzed, the target goal was reached for OH 2.2 only (Figure 1). This indicator’s target prevalence was 25.9% of children ages 6–9 years with dental decay. The baseline estimate was 28.8% (CI 95% 25.2–32.4) and the 2013–2016 estimate was 15.5% (CI 95% 13.2–18.0) (Table 2).

Demographic groups that saw a significant percent change in 6–9-year-olds with dental caries included non-Hispanic or Latino White children (−15.4%, *p* = 0.0482), those with private insurance (−11.35%, *p* = 0.0378), those with public insurance (−10.20%, *p* = 0.0408), and uninsured individuals (−8.38%, *p* = 0.0422) (Table 3).

Among adolescents aged 13–15 years old, the demographic groups that experienced a significant percent change in the proportion of individuals with dental caries included those with a family income of 200–399% of the federal poverty guideline (−14.37%, *p* = 0.0358), and 400–499% of the federal poverty guideline (−27.54%, *p* = 0.0032), as well as privately (−11.78%, *p* = 0.0495) and publicly (−11.23%, *p* = 0.0482) insured individuals (Table 4).

All demographic categories considered for OH 2.2 saw a significant decrease. Non-Hispanic or Latino White individuals experienced the greatest decrease (−55.56%, *p* = 0.0013), while non-Hispanic or Latino Black individuals experienced the smallest percent decrease (−36.69%, *p* = 0.0427) (Table 2). When comparing across insurance statuses, a disparity in the degree of change was observed between publicly insured individuals (−4.66 %, *p* = 0.0015) who had the greatest decrease compared to uninsured individuals (−14.25%, *p* = 0.0648), who saw the smallest percent change of the insurance categories considered (Table 2). Worth noting is that the proportion estimate for the uninsured demographic group had greater variability compared to other groups (CI 95% 21.6–45.6), but nonetheless remains the insurance category that had the greatest proportion of 6–9-year-olds with dental decay (Table 2).

Among adults 35–44 years old, the proportion of individuals with dental decay increased significantly for those with a family income 500% of the federal poverty guideline (+40.74%, *p* = 0.0258) (Table 5).

A significant decrease in the proportion of adults 45–64 years old with a permanent tooth extraction was observed for individuals with family incomes of the following percentages of the federal poverty guidelines: 100–199% (010.23%, *p* = 0.0249), 400–499% (−15.96%, *p* = 0.0143), and 500% (−13.34%, *p* = 0.01870) (Table 6). A 9.20% decrease was observed for publicly insured individuals (*p* = 0.0648) (Table 6).

## 4. Discussion

### 4.1. Context and Strength of This Study

This study critically evaluates progress in reducing oral health disparities using selected Healthy People 2020 indicators. There is a scarcity of research in the field of oral health, social epidemiology, and dental public health, reflecting the fact that comprehensive oral health surveillance programs were developed relatively recently. The National Oral Health Surveillance System, for instance, was introduced to track progress toward the Healthy People 2010 goals, as state-level oral health survey data was scarce prior to the turn of the century [8]. From a literature search of keywords “Healthy People 2020” AND “Oral Health” on PubMed (searched on 20 August 2021), only one 2012 study by Dye et al. specifically assessed oral health disparities with respect to the Healthy People 2020 objectives. The study, moreover, offers an assessment of the indicators as of 2010, eliciting a need for a follow-up evaluation of the indicators towards the end of the decade [9]. Worth noting is the fact that the Oral Health Surveillance Report issued by the CDC in 2019 published estimates based on NHANES data for a baseline period of 1994–2004 and a follow-up of 2011–2016. This report, however, provides singular total estimates without elucidating the nuances and variations in estimates across time periods for individual sociodemographic strata. In keeping with the goal of understanding not only progress towards the Healthy People 2020 goals but also disparities in progress, our study aligns with a call for constant assessment of Healthy People 2020 oral health objectives, placing our findings in the context of social determinants of health in the present day.

Taken together, there is evidence to suggest that progress has been made towards reducing oral health disparities in the United States, most notably in the goal of reducing the proportion of children with untreated dental decay. Our findings corroborate the trends described in *Oral Health in America*, which reports that untreated tooth decay has significantly decreased for children under 12, with the largest decrease occurring in children aged 2–5 years [3]. Factors contributing to this progress could include increased awareness of the positive association between sugar consumption and prevalence of dental decay in children over the past decade [10]. While school-based dental sealant programs were found to promote accessibility for a subgroup of Black students, overall oral health outcomes were still lacking compared to White students particularly when insurance status was confounded with race [11,12]. When comparing outcomes in children from similar socioeconomic backgrounds who were both covered by private insurance, Black children were found to receive fewer preventative dental procedures [12]. This trend could explain our finding that the most recent prevalence estimates of dental decay in children ages 6–9 showed a significant difference from 12.1% in White children to 22.6% in Black children. Despite OH 2.2 seeing the greatest achievement towards the 2013–2016 target, analysis at the level of racial substrata indicates that disparities are still present. These findings are consistent with previous studies in U.S. schoolchildren, showing suboptimal states of overall dental health in almost ⅓ of Hispanic and non-White children compared to less than ¼ in White children [13]. Interestingly, *Oral Health in America* identifies decreases in the prevalence of untreated dental caries as benefiting children from minority racial backgrounds the most, while income level remains the demographic characteristic to have the most disparities [3]. The data for our analysis of OH 2.2 among higher income levels did not meet thresholds for statistical significance, making comparisons with lower income levels unclear. However, it would be of interest to investigate whether the racial disparities we report are the product of intersectionality with income groups identified in the Surgeon General’s report. The proximate causes for these disparities are often factors such as socioeconomic standing and insurance status, which are disparate along racial strata [13,14]. Institutional factors such as socioeconomic barriers are aligned with one of three modes described by Williams and Collins with regards to the role of race in disadvantaging minority health status, the other two being cultural and individual-level racism [15].

The goals of reducing the proportion of adults with untreated dental decay and permanent tooth extractions require more attention moving into the next decade. No centralized means exists to provide healthcare to adults in the way that school-based interventions do for children. An emphasis on individual responsibility over community-based intervention presents particular barriers to adults who experience social limitations to their ability to make decisions about oral healthcare usage. Our findings show that as of 2013–2016, the prevalence of untreated dental decay in adults ages 35–44 varied from 42.6% (CI 95% 34.8–50.7) in Mexican American adults and 39.6% (CI 95% 32.9–46.8) in Black or African American adults to 24.3% (CI 95% 19.1–30.5) in White adults. It is critical to note the variability of these measures specifically in the range between the lower estimate for White adults compared to the upper estimate for Mexican American adults. Measures of spread reflect the extent of this disparity. These disparate estimates could be explained by low utilization of dental care, which is often a function of lower perceived need among disadvantaged racial minorities, such as Mexican American and Black or African American populations [16]. Studies of dental care behaviors among Black Americans from different age groups and localities indicate a perception of dental health as a low priority [12]. This trend could be explained by limited provider cultural competence and understandable skepticism from Black communities, who have been historically marginalized [12]. Lack of insurance may pose another major barrier to access for older adults since Medicare covers a relatively narrow range of essential medical procedures and Medicaid programs do not cover dental health for adults [16]. Considering that insurance status and socioeconomic standing have been known to intersect with race, this could present a valuable area for further analysis in the future.

One of the most pronounced trends across all five indicators was the positive correlation between income level and optimal oral health outcomes. Insured individuals were also more likely to experience improvement on each of the health metrics compared to uninsured individuals, and White Americans experienced more positive outcomes compared to Black and Hispanic Americans. These trends are not necessarily surprising, given the well-documented history of oral health disparities in the United States. Consistent with our reporting, the CDC identifies non-Hispanic Blacks, Hispanics, and American Indians and Alaska Natives as having the poorest oral health out of all other racial and ethnic groups in the United States [17]. Our findings are instead meaningful in underscoring the lasting impact of disparities even despite the progress that has been made over the past decade. They highlight subgroups in the American population that should be the focus of future interventions to foster more comprehensive and equitable healthcare.

### 4.2. Study Limitations and Strengths

Limitations of this study include its use of secondary data. Responses from the National Health and Nutrition Examination Survey (NHANES) were used to obtain prevalence estimates rather than self-administered survey data. As a result, population demographic information was not as readily available, and our findings could not rely on self-controlled sampling methods. Another limiting factor was the availability of data. Prevalence estimates were not available for certain racial/ethnic groups and income levels. A request to the CDC for raw data and individual data was unfulfilled due to the unavailability of these datasets. There were also limitations to the availability of punctual and/or interval statistics including mean, median, standard deviation, and confidence intervals for parameters used, as these measures were already calculated in the secondary datasets, making more precise analyses difficult. This study therefore analyzes aggregate data rather than individual survey responses. The possibility of ecological fallacy thereby limits the extent to which our findings are representative of the current distribution of selected indicators on individual, county, and state levels.

The descriptive aspect of this study is advantageous given the scarcity of similar research in oral health epidemiology. A descriptive analysis of the indicators therefore provides a useful basis for future studies that can further assess the effectiveness of and progress towards Healthy People’s goals.

### 4.3. Policy Implications and Future Directions

Future directions should address the intersectional nature of social determinants of health. Though it was not in the scope of this study’s aims to parse out intersections of race, socioeconomic status, income, and other variables’ combined effects on health outcomes, the data suggest that progress is limited for groups that are more socially vulnerable within distinct classifications. It would therefore be of interest for future studies to consider whether these trends are the product of intersectionality. One primary focus should be improving the equity of services offered by public oral healthcare [18]. Although public insurance covers similar services as private insurers, many oral healthcare benefits are considered optional rather than essential for eligible adults under the Affordable Care Act [19]. These services are therefore limited by state, making them inaccessible to a substantial proportion of eligible adults [18]. Thus, federal and state programs should seek to identify not only populations that are uninsured but also those that cannot take advantage of services despite being eligible for federal programs. Another primary area of consideration should address systemic racial inequities that continue to marginalize minority communities. Prioritizing cultural competence among healthcare practitioners is key because the power dynamics between racial groups are often projected onto the dynamic between dentists and patients, leading members of underserved or marginalized cultural backgrounds to minimize their contact with healthcare services [18]. Programs designed to reduce language barriers, link health education to cultural education, and localize health centers in communities with large Black, Hispanic, and Mexican American populations would be progressive directions. Furthermore, policy measures focusing on the efficacy of public insurance and oral health coverage can more effectively address concerns for older adult populations. In fact, as of October 2020, American Public Health Association’s Governing Council adopted new policies including a measure that would urge Congress to remove the exclusion of dental health benefits from Medicare and amend the ACA to identify adult oral health care as a necessity [20]. Such policy measures are progressive in addressing the concerns identified in this study.

More targeted sampling methods should also be used to fill in gaps from the NHANES survey data. Obtaining primary source records of usage from dental practices can address any limitations caused by survey response bias. Exploratory analyses based on secondary data, such as the present study, are also useful for improving the content validity of open access databases. Future studies can more closely analyze any particular social determinant on a smaller scale. The converse approach of taking a global perspective is another viable application for this study. The World Health Organization’s Global Goals for Oral Health in 2020 outline objectives that share much overlap with Healthy People’s indicators [21]. To address the broad Sustainable Development Goal of promoting good health and wellbeing, the WHO’s indicators are useful tools to think of oral health on a global scale while enacting actionable measures on a local scale [21]. Appropriate use of this guiding document can be a powerful tool in working towards progressive oral health policies on a community scale in the United States. These directions will serve in devising more specific programs to promote accessibility and cultural competence as previously suggested. Most importantly, the findings from studies of progress over the past decade will serve in effective goal-setting for oral health indicators in the future.

## 5. Conclusions

This study provides an important assessment of progress towards selected oral health indicators as part of Healthy People 2020. Our results suggest that significant progress was made on indicators concerning the oral health of children, such as reducing the proportion of children with untreated dental decay. However, racial and socioeconomic disparities persist, and limited progress has been made for adult oral health indicators. Policies should aim to reduce disparities by implementing cultural health education training for professionals and instituting accessible translation services. At the local level, health centers should be localized in communities with large Black and Hispanic populations. At the federal level, policies should seek to eliminate the exclusion of dental health benefits from public insurance programs.

## Figures and Tables

**Figure 1 ijerph-19-05250-f001:**
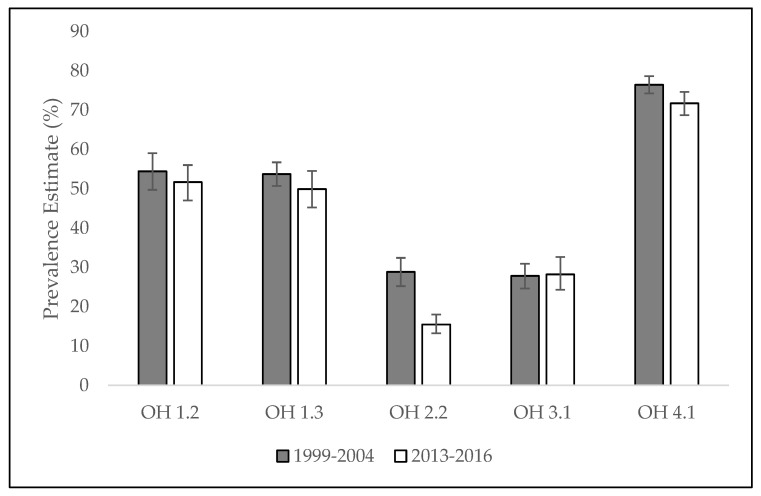
Proportion estimates for selected indicators. Error bars represent 95% CI. Healthy People 2020. USA, 1999–2004/ 2013–2016. **Source**: Healthy People 2020.

**Table 1 ijerph-19-05250-t001:** Description of the selected indicators, Healthy People 2020.

Indicator		Description	Calculation Method	
Group	Subgroup			
Oral health indicators of children and adolescents	OH 1.2: Caries 6–9 years old	Aims to reduce the proportion of children aged 6 to 9 years with dental caries in their deciduous or permanent teeth.	Number of children aged 6 to 9 years with coronary caries or presence of filling in at least one primary or permanent tooth or evidence of absent tooth due to caries.	×100
			Number of children aged 6 to 9 years with coronal caries codes valid for at least one primary or permanent tooth.	
	OH 1.3: Caries 13–15 years old	Aims to reduce the proportion of adolescents aged 13 to 15 years with dental caries in their permanent teeth.	Number of adolescents aged 13 to 15 years with coronary caries or presence of filling observed in at least one permanent tooth or evidence of a permanent tooth absent due to caries.	×100
			Number of adolescents aged 13 to 15 years with valid coronary caries codes for at least one permanent tooth.	
	OH 2.2: Decay 6–9 years old	Aims to reduce the proportion of children aged 6 to 9 years with untreated dental caries in their deciduous or permanent teeth.	Number of children aged 6 to 9 years with coronary caries that has not been restored observed in at least one deciduous or permanent tooth.	×100
			Number of children aged 6 to 9 years with at least one deciduous or permanent tooth present and valid coronary caries codes for at least one deciduous or permanent tooth.	
Oral health indicators of adults	OH 3.1: Decay 35–44 years old	Aims to reduce the proportion of adults 35 to 44 years old with untreated dental caries.	Number of adults 35 to 44 years with coronal caries that have not been restored in at least one permanent tooth.	×100
			Number of adults from 35 to 44 years with at least one permanent tooth present and valid codes of coronary caries for at least one permanent tooth.	
	OH 4.1: Permanent tooth extraction 45–64 years old	Aims to reduce the proportion of adults aged 45 to 64 years who have already had a permanent tooth extracted by dental caries or periodontal disease.	Number of people aged 45 to 64 years with valid codes for 28 permanent teeth exclusive to third molars.	×100
			Number of people aged 45 to 64 years with a clinical confirmation of less than 28 natural teeth present (tooth loss by caries or periodontal disease), exclusive to third molars.	
**Source**: Census. Data collection was conducted in December of the year 2019.				

**Table 2 ijerph-19-05250-t002:** Distribution of OH 2.2: decay 6–9 years, according to sociodemographic characteristics. Healthy People 2020. USA, 1999–2004/2013–2016.

Variable	Category	1999–2004			2013–2016			Change (−46.18%)	*p*-Value ***
		Estimate (28.8%)	StdErr (1.765)	CI 95% (25.2–32.4)	Estimate 15.5	StdErr 1.181	CI 95% 13.2–18.0		
**Sex**	Male	30.1	2.126	25.8–34.4	15.9	1.336	13.3–18.8	**−47.18**	0.0294
	Female	27.4	2.688	22.0–32.9	15.0	1.709	11.8–18.8	**−45.26**	0.0214
**Race/Ethnicity**	Hispanic or Latino	—	—	—	—	—	—	—	—
	Not Hispanic or Latino	41.4	2.596	36.1–46.6	—	—	—	**−55.56**	0.0013
	Mexican American	—	—	—	18.4	2.409	14.0–23.9	—	—
	Black or African American only, not Hispanic or Latino	35.7	1.582	32.5–38.9	22.6	1.974	18.8–26.9	**−36.69**	0.0427
	White only, not Hispanic or Latino	25.1	2.762	19.5–30.7	12.1	1.943	8.7–16.7	**−51.79**	0.0066
**Income (% poverty** **guidelines)**	<100	41.3	2.275	36.7–45.9	23.5	2.257	19.2–28.4	**−43.10**	0.0048
	100–199	35.7	3.509	28.6–42.7	18.1	1.979	14.4–22.5	**−49.30**	0.0021
	200–399	22.8	2.537	17.6–27.9	13.4	2.054	9.7–18.2	**−41.23**	0.3248
	400–499	—	—	—	—	—	—	—	—
	500	11.4	3.286	4.6–18.2	—	—	—	—	—
**Country of Birth**	US	28.2	1.896	24.4–32.0	15.1	1.201	12.8–17.7	**−46.45**	0.0235
	Outside US	41.0	5.289	29.9–52.2	25.9	5.850	15.8–39.5	—	—
**Insurance Status**	Insured	27.4	1.581	24.2–30.6	14.5	1.247	12.2–17.3	**−47.08**	0.0248
	Private	21.8	1.933	17.9–25.7	11.5	1.597	8.6–15.2	**−47.25**	0.0433
	Uninsured	37.9	3.964	29.9–45.9	32.5	5.975	21.6–45.6	**−14.25**	0.0648
	Public	39.7	2.287	35.1–44.3	18.0	1.644	14.9–21.6	**−54.66**	0.0015
**Source**: Healthy People 2020									

* Binomial Test—significance level (*p*-value < 0.05). Note: (—) no information and/or data did not meet statistical reliability.

**Table 3 ijerph-19-05250-t003:** Distribution of OH 1.2: caries 6–9 years, according to sociodemographic characteristics. Healthy People 2020. USA, 1999–2004/2013–2016.

Variable	Category	1999–2004			2013–2016			Change (−5.15%)	*p*-Value ***
		Estimate (54.4%)	StdErr (2.303)	CI 95% (49.7–59.0)	Estimate 51.7	StdErr 2.208	CI 95% 47.0–56.0		
**Sex**	Male	56.8	2.431	51.9–61.7	53.4	2.674	47.9–58.8	**−5.99**	0.4785
	Female	51.7	3.138	45.4–58.1	49.6	2.909	43.7–55.5	**−4.06**	0.2458
**Race/Ethnicity**	Hispanic or Latino	—	—	—	—	—	—	**—**	**—**
	Not Hispanic or Latino	—	—	—	—	—	—	**—**	**—**
	Mexican American	70.3	2.228	65.8–74.9	72.5	3.088	65.8–78.4	**3.13**	0.4586
	Black or African American only, not Hispanic or Latino	55.6	1.982	51.5–59.6	53.5	2.141	49.1–57.8	**−3.78**	0.1780
	White only, not Hispanic or Latino	50.0	3.059	43.8–56.2	42.3	3.174	36.0–48.9	**−15.40**	**0.0428**
**Income (% poverty guidelines)**	<100	68.4	3.377	61.6–75.2	66.0	2.893	59.9–71.7	**−3.51**	0.3333
	100–199	62.7	3.241	56.2–69.3	59.0	2.769	53.2–64.5	**−5.90**	0.4871
	200–399	46.3	3.128	40.0–52.6	42.5	2.700	37.1–48.0	**−8.21**	0.2648
	400–499	44.1	6.715	29.5–58.8	44.0	5.597	32.4–56.3	**−0.23**	0.4387
	500	30.7	5.263	19.8–41.6	32.8	4.885	23.7–43.4	**6.84**	0.4395
**Country of Birth**	US	53.9	2.364	49.2–58.7	51.2	2.260	46.5–55.7	**−5.01**	0.3355
	Outside US	63.3	6.082	50.4–76.1	63.0	4.680	53.1–72.0	**−0.47**	0.4361
**Insurance Status**	Insured	53.9	2.384	49.1–58.7	51.3	2.267	46.6–55.9	**−4.82**	0.3810
	Private	46.7	2.772	41.1–52.3	41.4	2.823	35.8–47.3	**−11.35**	**0.0378**
	Uninsured	59.7	5.151	49.2–70.1	54.7	6.030	42.3–66.5	**−8.38**	**0.0422**
	Public	69.6	2.563	64.5–74.8	62.5	2.189	58.0–66.9	**−10.20**	**0.0408**
**Source**: Healthy People 2020									

* Binomial Test—significance level (*p*-value < 0.05). Note: (—) no information and/or data did not meet statistical reliability.

**Table 4 ijerph-19-05250-t004:** Distribution of OH 1.3: caries 13–15 years, according to sociodemographic characteristics. Healthy People 2020. USA, 1999–2004/2013–2016.

Variable	Category	1999–2004			2013–2016			Change (−7.08%)	*p*-Value ***
		Estimate (53.7%)	StdErr (1.483)	CI 95% (50.7–56.7)	Estimate 49.9	StdErr 2.288	CI 95% 45.2–54.5		
**Sex**	Male	50.3	2.325	45.6–55.0	49.7	3.355	42.9–56.6	**−1.19**	0.4458
	Female	57.2	1.930	53.3–61.1	50.0	2.912	44.1–55.9	**−12.59**	0.2578
**Race/Ethnicity**	Hispanic or Latino	—	—	—	—	—	—	**—**	**—**
	Not Hispanic or Latino	—	—	—	—	—	—	**—**	**—**
	Mexican American	62.1	2.115	57.8–66.5	64.0	4.784	53.7–73.1	**3.06**	0.4334
	Black or African American only, not Hispanic or Latino	48.1	2.242	43.5–52.7	49.6	3.881	41.8–57.5	**3.12**	0.4982
	White only, not Hispanic or Latino	52.4	2.404	47.6–57.3	45.6	3.099	39.4–52.0	**−12.98**	0.2498
**Income (% poverty** **guidelines)**	<100	61.8	2.329	57.1–66.5	60.7	4.430	51.4–69.3	**−1.78**	0.3468
	100–199	59.5	2.801	53.9–66.5	60.4	4.091	51.9–68.4	**1.51**	0.4268
	200–399	52.9	2.536	47.8–58.0	45.3	2.672	39.9–50.8	**−14.37**	**0.0358**
	400–499	51.2	4.068	42.9–59.5	37.1	9.608	20.3–57.8	**−27.54**	**0.0032**
	500	34.3	4.588	24.9–43.7	32.5	6.363	21.1–46.6	**−5.25**	0.2782
**Country of Birth**	US	52.9	1.535	49.8–56.0	49.0	2.456	44.0–54.0	**−7.37**	0.2517
	Outside US	63.0	4.552	53.7–72.4	63.2	7.104	48.0–76.3	**0.32**	0.3345
**Insurance Status**	Insured	52.7	1.639	49.4–56.0	48.8	2.340	44.1–53.6	**−7.40**	0.287
	Private	48.4	1.870	44.7–52.2	42.7	3.061	36.6–49.1	**−11.78**	**0.0495**
	Uninsured	59.9	3.950	51.9–67.9	63.8	5.862	51.2–74.8	**6.51**	0.2251
	Public	64.1	2.768	58.5–69.6	56.9	3.224	50.3–63.4	**−11.23**	**0.0482**
**Source**: Healthy People 2020									

* Binomial Test—significance level (*p*-value < 0.05). Note: (—) no information and/or data did not meet statistical reliability.

**Table 5 ijerph-19-05250-t005:** Distribution of OH 3.1: decay 35–44 years, according to sociodemographic characteristics. Healthy People 2020. USA, 1999–2004/2013–2016.

Variable	Category	1999–2004			2013–2016			Change (−1.44%)	*p*-Value ***
		Estimate (27.8%)	StdErr (1.574)	CI 95% (24.6–30.9)	Estimate 28.2	StdErr 2.047	CI 95% 24.3–32.6		
**Sex**	Male	30.1	2.068	25.9–34.2	30.0	2.796	24.6–36.0	**−0.33**	0.3490
	Female	25.4	1.601	22.2–28.6	26.7	1.762	23.2–30.4	**5.12**	0.2022
**Race/Ethnicity**	Hispanic or Latino	—	—	—	—	—	—	—	—
	Not Hispanic or Latino	—	—	—	—	—	—	—	—
	Mexican American	40.2	2.836	34.3–46.0	42.6	3.916	34.8–50.7	**5.97**	0.3458
	Black or African American only, not Hispanic or Latino	40.5	2.712	35.0–46.0	39.6	3.416	32.9–46.8	**−2.22**	0.4698
	White only, not Hispanic or Latino	22.8	2.077	18.6–27.0	24.3	2.793	19.1–30.5	**6.58**	0.2047
**Income (% poverty** **guidelines)**	<100	49.4	3.196	42.9–55.8	50.0	2.990	43.9–56.1	**1.21**	0.1789
	100–199	44.9	2.948	39.0–50.9	38.6	3.255	32.2–45.5	**−14.03**	0.2024
	200–399	27.1	2.354	22.4–31.9	28.2	2.648	23.1–33.9	**4.06**	0.2197
	400–499	21.0	3.106	14.7–27.4	14.2	3.144	8.9–21.9	**−32.38**	0.3412
	500	8.1	1.864	4.3–11.8	11.4	2.557	7.1–17.7	**40.74**	**0.0258**
**Country of Birth**	US	26.7	1.818	23.0–30.4	27.6	2.501	22.8–33.0	**3.37**	0.2784
	Outside US	32.4	2.922	26.5–38.4	30.0	2.560	25.1–35.5	**−7.41**	—
**Insurance Status**	Insured	23.1	1.855	19.3–26.8	22.4	1.814	19.0–26.4	**−3.03**	0.3574
	Private	20.0	1.691	16.6–23.4	18.5	1.893	15.0–22.7	**−7.50**	0.3111
	Uninsured	46.3	2.437	41.4–51.2	47.9	2.682	42.5–53.4	**3.46**	0.1758
	Public	49.7	6.245	37.0–62.3	40.1	2.722	34.7–45.7	**−19.32**	0.0587
**Source**: Healthy People 2020									

* Binomial Test—significance level (*p*-value < 0.05). Note: (—) no information and/or data did not meet statistical reliability.

**Table 6 ijerph-19-05250-t006:** Distribution of OH 4.1 permanent tooth extraction 45–64 years, according to sociodemographic characteristics. Healthy People 2020. USA, 1999–2004/2013–2016.

**Variable**	**Category**	**1999–2004**			**2013–2016**			**Change (−6.15%)**	***p*-Value *****
		**Estimate (76.4%)**	**StdErr (1.122)**	**CI 95% (74.2–78.6)**	**Estimate 71.7**	**StdErr 1.443**	**CI 95% 68.7–74.6**		
**Sex**	Male	75.3	1.336	72.7–78.0	71.5	1884	67.5–75.1	**−5.05**	0.3257
	Female	77.3	1.309	74.7–79.9	71.9	1.705	68.3–75.3	**−6.99**	0.1715
**Race/Ethnicity**	Hispanic or Latino	—	—	—	—	—	—	—	—
	Not Hispanic or Latino	—	—	—	—	—	—	—	—
	Mexican American	80.3	1.699	77.0–83.6	76.7	1902	72.6–80.4	**−4.48**	0.3644
	Black or African American only, not Hispanic or Latino	92.9	1	91.0–94.7	85.9	1302	83.0–88.3	**−7.53**	0.1243
	White only, not Hispanic or Latino	72.6	1.469	69.7–75.5	67.7	1878	63.7–71.4	**−6.75**	0.2241
**Income (% poverty** **guidelines)**	<100	89.9	2.083	85.8–94.0	91.0	1691	86.9–93.9	**1.22**	0.4248
	100–199	92.9	1.309	90.4–95.5	83.4	1794	79.4–86.7	**−10.23**	**0.0249**
	200–399	81.1	2.179	76.8–85.4	80.2	1859	76.2–83.8	**−1.11**	0.4557
	400–499	75.8	2.588	70.7–80.9	63.7	4639	53.8–72.5	**−15.96**	**0.0143**
	500	62.2	1.903	58.5–66.0	53.9	2752	48.2–59.4	**−13.34**	**0.1870**
**Country of Birth**	US	75.2	1.268	72.8–77.7	70.8	1607	67.5–74.0	**−5.85**	0.2543
	Outside US	83.8	2.141	79.0–88.0	76.4	1844	72.5–80.0	**−8.83**	0.0732
**Insurance Status**	Insured	74.2	1.220	71.9–76.6	69.1	1453	66.0–72.0	**−6.87**	0.2128
	Private	71.8	1.262	69.4–74.3	64.6	1675	61.2–68.0	**−10.03**	0.1874
	Uninsured	90.0	1.624	86.9–93.2	88.0	1415	84.8–90.6	**−2.22**	0.3245
	Public	93.5	1.303	90.9–96.0	84.9	2071	80.2–88.7	**−9.20**	**0.0648**
**Source**: Healthy People 2020									

* Binomial Test—significance level (*p*-value < 0.05). Note: (—) no information and/or data did not meet statistical reliability.

## Data Availability

The data presented in this study are openly available in the Healthy People 2020 repository, [https://www.healthypeople.gov/2020/data-search/Search-the-Data#topic-area=3511] (accessed on 1 December 2019).

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
