# Peer review of "Are We Getting Any Better? A Critical Analysis of Selected Healthy People 2020 Oral Health Indicators in 1999–2004 and 2013–2016, USA"

_ijerph, 2022, doi:10.3390/ijerph19095250_

Round 1

Reviewer 1 Report

The authors’ stated goal is to analyze oral health disparities in the US from the early 2000s to 2013-14 according to sociodemographic characteristics, by using NHANES data from two cycles. The authors refer to Healthy People 2020 as the source of five oral health indicators (e.g., proportion of children ages 6-9 years with caries experience).  They perform descriptive analyses regarding proportion of people experiencing each oral health indicator. They also estimate differences in each proportion, by sociodemographic subgroups, between the two NHANES cycles. They perform statistical tests of these differences.

I agree with the authors about the importance of this topic and about the urgency to address these extreme health disparities. Nevertheless, I have concerns about the approach and write-up.

The authors focus much too strongly on results of dichotomous significance testing and too little on point estimates, their direction, and their variability. For the differences between time periods, 95% CIs would be vastly preferred to conclusions based on dichotomous significance thresholds, as has been well articulated by the American Statistical Association (https://www.amstat.org/asa/files/pdfs/p-valuestatement.pdf, https://www.tandfonline.com/doi/full/10.1080/00031305.2016.1154108, https://www.tandfonline.com/doi/full/10.1080/00031305.2019.1583913 ).

For example, in the results referring to proportion of children with caries experience by ages 6-9 years, the authors highlight only the results meeting statistical significance. Yet there were income groups in which big changes occurred. The authors should report estimated differences and their variability rather than only results passing threshold values. Also by focusing only on statistical significance, the authors miss the opportunity to highlight in which groups the proportion with caries increased or decreased. Rarely (if ever) is statistical significance more important to report than the direction of a difference.

Hampering my ability to read and understand the manuscript was that the table formatting is unusual and somewhat sloppy. There are horizontal lines in seemingly random places, and the arbitrary placement of the “total” change parenthetically in the row headings makes the tables confusing. For each time period, the authors present two nearly redundant measures of variability for the proportion (the standard error and the 95% confidence interval – which can be estimated roughly as the estimate +/- 1.96*SE), whereas for the difference between time period they present neither. 95% CIs are important to provide and probably more useful than the SE.

I agree with the authors that diverging trends can be an important indicator of a disparity, but there seemed to be a disconnect between their goals and discussion versus the type of analysis they conducted. For example, just as important as differences in trends are differences in absolute disparities (in the most recent data possible), or differences in proportion meeting the HP 2020 goals by sociodemographic category. The authors make strong points about intersectionality (with which I happen to agree), but they do not examine whether disparities in poor oral health (or in oral health trends) are worse among people who are socially vulnerable in more than one dimension.

I believe there are more recent NHANES data that should be included in this study. And, some of the same results presented here are presented in a report from the CDC (https://www.cdc.gov/oralhealth/publications/OHSR-2019-list-of-tables.html), with data up through 2016. The authors should acknowledge the more recent data and any published reports or discussions.

Discussion: I’m not sure this study is the first to critically evaluate progress in reducing oral health disparities using Healthy People 2020 indicators; certainly others have commented on the failure to meet Healthy People 2020 goals in sociodemographically or economically vulnerable groups (e.g., https://www.ncbi.nlm.nih.gov/pmc/articles/PMC5858650/).

Reviewer 2 Report

The paper "Are We Getting Any Better? A Critical Analysis of Selected 2 Healthy People 2020 Oral Health Indicators in 1999-2004 and 3 2013-2014, USA." represents an interesting analysis of oral health disparities in correlation to socio-demographic parameters.  The study is clearly described, and the only question that would need clarification concerns the description of the study: How and why were the 2 time periods (1999-2004 and 2013-2014) selected for the analysis? Also, why were the mentioned oral health indicators selected? Maybe a more comprehensive description of the Healthy People initiative would give more insight.   

Reviewer 3 Report

This study aims to assess the progress in dental cavities for Healthy People 2020 from 1999-2004 to 2013-2016. some suggestions for the authors:

1) The data was extracted in 2020 and needs to be updated. Why not use more recent data after 2016?  The title is confusing as only dental cavities indicators were assessed

2) The introduction should be more focused. Not sure why they introduced the intersectional perspective? As it is written, it is hard to justify the study.

3) The first row in the tables is confusing. A bar chart might be more informative

4) May need to include absolute change in the table. The current change column is relative change? Need to describe how this is calculated. 

6) Please compare the results with existing literature. NIH just released a report "Oral Health in America" which summarized major achievements since 2000.
